# Sexist, Racist, and Homophobic Violence against Paramedics in a Single Canadian Site

**DOI:** 10.3390/ijerph21040505

**Published:** 2024-04-19

**Authors:** Justin Mausz, Joel D’Eath, Nicholas A. Jackson, Mandy Johnston, Alan M. Batt, Elizabeth A. Donnelly

**Affiliations:** 1Peel Regional Paramedic Services, 1600 Bovaird Drive East, Brampton, ON L6V 4R5, Canada; joel.death@peelregion.ca (J.D.); nicholas.jackson@peelregion.ca (N.A.J.); mandy.johnston@peelregion.ca (M.J.); 2Department of Family and Community Medicine, Temerty Faculty of Medicine, University of Toronto, 500 University Avenue, Toronto, ON M5G 1V7, Canada; 3Faculty of Health Sciences, Queen’s University, 99 University Avenue, Kingston, ON K7L 3N6, Canada; alan.batt@queensu.ca; 4Department of Paramedicine, Monash University, Peninsula Campus, Building H, 47-49 Moorooduc Hwy, Frankston, VIC 3199, Australia; 5School of Social Work, University of Windsor, 167 Ferry Street, Room 213, Windsor, ON N9A 0C5, Canada; donnelly@uwindsor.ca

**Keywords:** paramedics, emergency medical services, emergency medical technicians, violence, racism, sexism, homophobia

## Abstract

Violence against paramedics is widely recognized as a serious, but underreported, problem. While injurious physical attacks on paramedics are generally reported, non-physical violence is less likely to be documented. Verbal abuse can be very distressing, particularly if the harassment targets personal or cultural identities, such as race, ethnicity, gender, or sexual orientation. Leveraging a novel, point-of-event reporting process, our objective was to estimate the prevalence of harassment on identity grounds against paramedics in a single paramedic service in Ontario, Canada, and assess its potentially differential impact on emotional distress. In an analysis of 502 reports filed between 1 February 2021 and 28 February 2022, two paramedic supervisors independently coded the free-text narrative descriptions of violent encounters for themes suggestive of sexism, racism, and homophobia. We achieved high inter-rater agreement across the dimensions (k = 0.73–0.83), and after resolving discrepant cases, we found that one in four violent reports documented abuse on at least one of the identity grounds. In these cases, paramedics were 60% more likely to indicate being emotionally distressed than for other forms of violence. Our findings offer unique insight into the type of vitriol paramedics experience over the course of their work and its potential for psychological harm.

## 1. Introduction

### 1.1. Violence against Paramedics

Violence is widely recognized as a serious occupational health issue affecting paramedics, having the potential for significant physical and psychological harm [1]. For example, data from the United States (US) Bureau of Labor Statistics illustrate that paramedics experience a risk of injury from violence that is six times greater than the US population and 60% greater than comparable health professions [2]. When surveyed, a concerning majority of paramedics indicated having experienced various forms of violence—including physical and sexual assault—within the past year or throughout their careers [3,4,5,6,7,8,9,10]. The research parallels growing reports in the media of paramedics being seriously injured (and sometimes killed) after violent attacks by members of the public [11,12,13,14,15,16]. While sentinel events tend to be documented in injury statistics or press coverage, they occur against a backdrop of pervasive institutional under-reporting [1,17,18]. For example, when Bigham and colleagues surveyed paramedics from two Canadian provinces, despite more than 75% of participants indicating past-year exposure to violence, less than 20% documented the incidents or reported the encounters to police or supervisors [3].

### 1.2. Under-Reporting

In an earlier study, we identified that the organizational culture within paramedicine appears to sustain norms that limit reporting by implicitly positioning the ability of paramedics to ‘brush off’ incidents of violence as an expected professional competency [18]. Within this construction, while injurious encounters may be documented, incidents perceived as less serious—but more prevalent—are seen as unpreventable and (hence) ‘not worth’ reporting [18]. Under-reporting has been highlighted in the literature as a significant problem [17], one that severely limits researchers’ and policymakers’ understanding of both the scope and type of violence that paramedics experience in their work.

### 1.3. Non-Physical Violence

Although violence against paramedics can (and often does) take the form of physical or sexual assault, pervasive under-reporting means that non-physical violence, such as threats, harassment, and verbal abuse, may be even more prevalent. Returning here to Bigham and colleagues’ 2014 survey, 14% of participants indicated that they had been sexually harassed, 41% had been threatened, and 67% had been subjected to verbal abuse in the past year [3]. Even in the absence of physical assault, verbal abuse can be distressing in its own right, especially if the abuse targets personal or cultural identities, such as race, gender, or sexual orientation. Previous research among healthcare workers has identified that harassment from patients on these grounds is both widespread and contributes to insomnia, depression, anxiety, burnout, attrition from healthcare, and suicide [19,20]—problems that have increased since the onset of the COVID-19 pandemic [21,22,23,24]. Unfortunately, the problem of under-reporting [17] means that there is a lack of similar evidence on non-physical violence against paramedics, creating a gap in research with potentially significant implications for the health and retention of the workforce. There is, then, a compelling need to study and respond to all forms of violence against paramedics, particularly those that are the least likely to be documented or reported.

### 1.4. The External Violence Incident Report

Our team developed a novel violence reporting process intended to overcome many of the administrative and organizational cultural barriers to reporting violence against paramedics. While the specifics have been reported in an earlier publication [25], in brief, the External Violence Incident Report (EVIR) was derived through an extensive stakeholder consultation and pilot testing process. The form is embedded in the electronic Patient Care Record (ePCR) and is designed to capture comprehensive quantitative and qualitative data about violent encounters at the time of events, as documented by the affected paramedic(s). Particularly relevant to this study, the form contains a free-text box where paramedics can type a detailed description of violent incidents without any restrictions on length. This allows for a novel study of this problem in that detailed, event-level descriptions of violent encounters can now be captured—a useful development in the field, as previous research on this subject has relied mostly on survey methods [26] or occupational injury statistics [2].

Qualitatively analyzing the free-text narrative descriptions of violent encounters may offer unique insight into the nature of the violence paramedics encounter in their work, including potential abuse on identity grounds such as race, ethnicity, gender, and sexual orientation—a valuable contribution to scholarship on this topic, given the gap that we have described. Therefore, our objectives were to (1) estimate the prevalence of (potentially) hate-motivated violence against paramedics, including sexism, misogyny, racism, and homophobia; and (2) assess the differential impact of hate-motivated violence on self-reported emotional distress.

## 2. Methods

### 2.1. Overview and Setting

This study is part of a broader research program, and a detailed description of the approach is provided in an earlier publication [27]. For this study, we analyzed the free-text narrative descriptions of violent encounters documented in a thirteen month subset of EVIRs following the introduction of the new reporting process.

Our research is situated in the Region of Peel in Ontario, Canada. Peel Regional Paramedic Services (PRPS) is the sole provider of publicly funded land ambulance and paramedic services to the municipalities of Brampton, Mississauga, and Caledon. PRPS employ approximately 750 primary and advanced care paramedics and 60 members in various leadership or administrative capacities, who also maintain certification to practice. The service provides coverage to a mixed urban and rural geography of 1200 km^2^ with a population of 1.38 million residents, responding to approximately 130,000 emergency calls per year. This positions the service as the second largest in the province by staffing and caseload.

The introduction of the EVIR occurred as part of a broader violence prevention program within the service that included crisis prevention and de-escalation training, new patient restraint equipment, revised safety procedures, and a public position statement of ‘zero tolerance’ for violence against paramedics.

### 2.2. Data Collection

The EVIR (available in the Appendix A) is a web-based form embedded in the electronic Patient Care Record alongside other documentation, and it includes a free-text box where paramedics can document a detailed narrative description of a violent encounter. The form imposes no word or character limits, and paramedics are encouraged to be both detailed and specific, incorporating the use of direct quotes where possible. When completing an EVIR, the documenting paramedic can also indicate whether they were physically harmed or ‘emotionally impacted’ (or both) at the time of reporting.

Provincial documentation standards require paramedics to complete an Ambulance Call Report (ACR) after every patient encounter, and they additionally require paramedics to complete an incident report in unusual circumstances. The EVIR is considered an incident report under these standards, and service policy instructs paramedics to complete an EVIR if they experienced verbal abuse, threats, sexual harassment, or physical or sexual assault during a call. Paramedics are additionally reminded to complete an EVIR when filing an ACR if they experienced violence during the call.

Our study window included a thirteen-month period following the introduction of the EVIR from 1 February 2021 to 28 February 2022. We included all violence reports filed during this period unless the documenting paramedic ‘opted out’ of secondary use of the form for research purposes.

### 2.3. Measures and Outcomes

Our primary outcome was the proportion of EVIRs that described violence (particularly verbal abuse) with racist, homophobic, sexist, or misogynistic connotations based on the description provided by the documenting paramedic. The Ontario Human Rights Code explains that all persons are protected under law from discrimination or harassment on the basis of several *protected grounds*, including sexual orientation, gender identity or expression; and race and related grounds, among others [28]. In addition, occupational health and safety legislation in Ontario protects workers (including paramedics) from harassment, which is defined in the Occupational Health and Safety Act (OHSA) as “engaging in a course of vexatious comment or conduct against a worker in a workplace that is known or ought reasonably to be known to be unwelcome” [29]. The purviews of both the Ontario Human Rights Code and the OHSA overlap, particularly in employment contexts, and we used both pieces of legislation as orienting concepts for identifying potentially hate-motivated violence in the EVIRs. Where an EVIR included hateful or offensive language, we referenced the wiki-style website www.urbandictionary.com to source definitions of the word grounded in contemporary usage to assess whether it could reasonably be construed as sexist or racist (for example). Importantly, EVIRs do not collect demographic information about the documenting paramedic, meaning that our assessment of sexism (for example) was focused specifically on the language itself rather than the degree to which the abuse was personally targeted.

Our secondary outcome was self-reported emotional distress, as documented by the reporting paramedic. The EVIR asks paramedics to indicate whether they are emotionally impacted at the time of the event, with response options for ‘yes’, ‘no’, and ‘I am uncertain’. We considered ‘yes’ a positive screen for emotional distress.

### 2.4. Analysis

We analyzed our data two ways: first, we decoupled the free-text narrative descriptions of violent encounters from all EVIRs included in this study and subjected the narratives to a structured form of qualitative content analysis [30]. For this task, we recruited two paramedic supervisors with knowledge of the violence prevention project and the relevant legislation to code each narrative in response to the prompt ‘does the narrative suggest violence on the basis of gender, sexual orientation, or race/ethnicity?’ Following an initial period of calibration where the two investigators coded a sample of 15 EVIRs with the principal investigator, the remaining narratives were coded independently. We then calculated the inter-rater agreement using a Kappa statistic and met as a group to resolve discrepant cases through consensus. For our secondary objective, we used chi-square tests to assess group differences in self-reported emotional distress between hate-motivated vs. non-hate motivated violence. All analyses were completed using SPSS Statistics (IBM Corporation, Armonk, NY, USA; version 28), and we followed convention in accepting a *p*-value of <0.05 with confidence intervals that exclude the null value to indicate statistical significance.

## 3. Results

### 3.1. Note to Readers

Our results include descriptions of sexist, racist, and homophobic comments that may be upsetting to some readers.

### 3.2. Overview

Between 1 February 2021 and 28 February 2022, a total of 722 paramedics in Peel Region responded to 119,295 emergency calls, and 563 EVIRs were generated. In 36 cases, the documenting paramedics did not consent to use of the form for research and the reports were excluded. A further 24 reports were missing a free-text narrative description and were excluded. Finally, one report was a duplicate of the same incident, and it was also excluded. This left 502 unique reports filed by a total of 266 paramedics for analysis (see Figure 1).

Our reviewers achieved substantial inter-rater agreement [31] for sexual orientation (k = 0.73), race (k = 0.76), and gender (k = 0.83; see Table 1), with violence on the basis of gender having the highest proportion of complete agreement between raters (78%) and race having the lowest (65%).

After resolving discrepant cases (Table 2), 18% of EVIRs were found to describe violence against paramedics on the basis of gender, 9% because of race (or related grounds), and 2% because of sexual orientation, with 25% documenting some form of abuse on any of the three protected grounds. Among the paramedics reporting violence on protected identity grounds (*N* = 96), the average number of reports was 1.32 (standard deviation [SD] 0.76), with a maximum per-paramedic frequency of six reports. This corresponds to 13% of the active-duty paramedic workforce reporting exposure to harassment on protected identity grounds during the study period.

### 3.3. Sexist and Misogynistic Comments

The paramedics documented a wide variety of incidents that our reviewers identified as having sexist or misogynistic connotations or undertones. These included, for example, calling paramedics offensive names (most commonly variations of ‘bitch’ or ‘cunt’) that our raters felt—in the context of the narrative descriptions—were inherently gendered. For example, one report described a (male) patient calling the paramedic a ‘fucking little bitch’ for asking the patient to wear a surgical mask during treatment.

We also observed several instances where the reports describe sexual language used (primarily) to demean women. For example, one report describes a patient commenting to a paramedic that (she) must have ‘sucked a lot of dick’ to get this job. Other reports document patients calling paramedics variations of ‘skank’, ‘whore’, or ‘slut’—often in combination with references to the paramedic’s breasts, buttocks, or jewelry (i.e., ‘eyebrow ring slut’). Much of the language was sexually demeaning, telling—when this information was documented—female paramedics to ‘suck my dick’ or ‘lick my balls’ in response to routine clinical interview questions. One report described a patient telling a (female) paramedic that she could ‘choke on a big Black cock and die’. We also identified several cases where male patients exposed themselves to female paramedics and masturbated in front of them.

Our raters also identified several cases where patients (or others at the scene) would refuse to speak to female paramedics, preferring instead to speak to males. For example, one report described a patient speaking only to the male paramedic, despite the female paramedic having the lead role on the call. Per the report, when the female paramedic tried to interview the patient, the patient told her to ‘shut the fuck up, bitch’.

### 3.4. Racist Comments

Our reviewers identified a wide variety of vitriolic comments documented in the EVIRs that were felt to convey racist overtones—either overtly, through the use of anti-Black or anti-Asian slurs, or in veiled references to racialized persons (e.g., ‘I hate all you people’). Here, again, although the EVIR does not collect demographic information about the reporting paramedic, we did observe cases where—in describing the incident—the documenting paramedic would indicate that the patient made references to their actual (or presumed) ethnicity. This included, for example, situations where patients would tell the paramedics to ‘go back where you came from, you fucking mutts’, call them various slurs in Punjabi or Korean, or utter threats that included an ethnic component (e.g., ‘I am going to kick your [racial slur] ass’). In some cases, the reports described racist sentiments that were expressed to other (presumably white) paramedics present at the scene (e.g., ‘I will get his [racial slur] ass’—referring to a racialized paramedic).

### 3.5. Homophobic Comments

Although not as common as sexist or racist sentiments in the reports, our reviewers identified a non-insignificant portion of EVIRs that documented homophobic language and slurs. Usually, the reports did not go into detail beyond ‘homophobic slurs’, but when the paramedics did include quotes, the most common were variations of the word ‘faggot’, and they were often uttered in combination with explicit or veiled threats: ‘You faggot, I am going to kill you’ or ‘It’s a good thing you’re not fags, I would have fun with you’. Less commonly, (presumably) female paramedics were called ‘carpet munching cunts’.

### 3.6. Impact on Paramedics

Of the 502 reports included in this study, 124 reports (25%) indicated that the documenting paramedic (*N* = 89) was ‘emotionally impacted’ by the incident at the time of reporting. Compared to other types of violence, incidents involving abuse on protected identity grounds were associated with an increased risk of emotional distress (31% vs. 22% of reports, odds ratio [OR] 1.59, 95% CI 1.01–2.48, *p* = 0.04).

## 4. Discussion

Our objective was to assess the frequency with which paramedics in our setting reported being subjected to sexist, misogynistic, racist, or homophobic abuse in the course of their work over a 13-month period. All told, we found that one in four reports filed by 96 paramedics documented some form of verbal abuse or harassment on protected identity grounds during the study, corresponding to 13% (≈1 in 8) of the active-duty work force. Averaged over the study period, this corresponds to a rate of one report of harassment on protected identity grounds every 3 days.

Given the historical under-reporting on this topic, ascertaining the scope and complexity of the problem of violence against paramedics has been challenging. Previous research has tended to rely on either self-report survey methods [3,4,5,6,7,18,32,33] or retrospective reviews of injury statistics databases [2,34,35]. While undeniably valuable, both approaches lack the granularity and specificity afforded by event-level data, and neither approach (to our knowledge) has specifically sought to estimate the prevalence of harassment on identity grounds. In that respect, our findings help to fill an important gap in the research by contributing unique insight into the texture of the abuse that paramedics encounter in the course of their work.

The research program to which this study is attached is intended to construct an epidemiological profile of violence against paramedics as an occupational health issue. From a research perspective, however, important questions remain unanswered. For example, future research should seek to quantify the impacts of incidental and recurrent exposure to workplace violence on health and well-being—including the impact of vitriolic verbal abuse and harassment. Using longitudinal or repeated sampling approaches to correlate violent encounters with self-report symptom measures for depression, anxiety, post-traumatic stress disorder, and burnout would be a welcome contribution to scholarship in this space, while also painting a clearer picture of the potential long-term harms from violence. Future research should also seek to find and evaluate potential risk mitigation strategies, such as flagging individuals with a history of violent behavior or providing de-escalation training to paramedics—both of which have been proposed as potential solutions to this problem [7,17,36,37,38]. Finally, researchers should engage paramedics to ascertain what forms of post-incident support they find most useful in mitigating the acute stress that is likely associated with being exposed to workplace violence. Similar work using qualitative interviews and focus groups of paramedics has been performed with good effect to develop supportive interventions after critical incidents, for example [39,40].

From a policy perspective, our findings have important implications amid what is increasingly recognized as a health human resource crisis that confronts healthcare broadly. There is now abundant research illustrating that healthcare providers are exposed to high rates of workplace violence and that, during the COVID-19 pandemic in particular, the violence increased [21,22,23,24,41]. In recent years, growing political and social divisions fueled by mis- or disinformation on public health measures has given rise to antipathy and hostility toward healthcare as an institution [22,42]. The result has been an increase in the harassment of healthcare workers, both online and in clinical settings [24], which, in turn, has led to burnout, depression, anxiety, and what has been described as an ‘exodus’ of healthcare workers exiting the professions [43,44]. These issues exist in paramedic services as well, with media reports describing staffing shortages and low morale as contributing to increasing wait times for ambulance calls [45,46,47]. In our province, the Ontario Association of Paramedic Chiefs has forecasted a shortage of 400 paramedics per year, a shortfall the provincial government is attempting to address [48]. Although not the focus of this study, it is reasonable to expect that violence may be a contributing factor to staffing shortages in paramedic services. There is a myriad of recommendations for workplace violence-prevention strategies in healthcare settings broadly, [49] but—to our knowledge—no such policy documents are specifically designed for the paramedic context. As a healthcare setting, the out-of-hospital context presents many unique challenges that potentially increase the risk of harm from violence, including limited resources, undifferentiated patients, ready access to weapons, and poorly controlled (or uncontrollable) scene environments, to name but a few. Provincial or national strategies for violence prevention policy alongside robust post-incident support would be welcome contributions.

Finally, it is worth noting that verbal abuse targeting individuals in protected groups is not novel or unique to paramedicine. Rather, it is evidence of the continued impact of the longstanding systems of injustice extant within our larger sociopolitical context. Our findings add to a substantial body of research demonstrating that systems of injustice continue to actively impact vulnerable groups and perpetuate inequality [50].

### Limitations

Our findings should be interpreted within the context of certain limitations. First, we must acknowledge that the violence reporting process was not intended to document harassment on protected identity grounds. Instead, we relied on the narrative description provided by documenting paramedics, with a particular emphasis on reports that included quotes from the harassers. Although paramedics are encouraged to be specific in their documentation and to include quotes, we nevertheless acknowledge our analysis required a value judgment on the part of our research team as to what constitutes sexist, racist, or homophobic language. Our raters were not racialized individuals themselves. In applying a conservative analytical frame within a context of under-reporting, this means that the actual frequencies at which paramedics are exposed to this vitriol is likely much higher. Second, our assessment of emotional distress is constrained to a single question on the reporting form answered at the time of the incident. In accepting only “yes” answers and excluding the “I am uncertain” responses, we were again applying a conservative analytical approach in our estimates. It is also worth noting that emotional distress may well manifest after some time has passed and may be worsened by repeated exposure, as is the case with several of our participants who reported multiple incidents of abuse on protected identity grounds. Again, our assessment of the potential psychological impacts of the abuse underestimates the true degree of harm experienced by the paramedics. Third, as mentioned earlier, the EVIR does not capture detailed demographic information for either perpetrators or the documenting paramedics. This meant that, unless the documenting paramedic specifically provided this information, we were unable to evaluate the directionality or differential targeting of potentially hate-motivated abuse (e.g., misogynistic comments from men *towards* women); rather, the focus of analysis was on whether the description documented abuse with misogynistic overtones. Finally, we took the paramedics’ accounts at face value, without attempting to ascertain the veracity of the claims—although we have no reason to doubt them.

## 5. Conclusions

Leveraging a novel, point-of-event incident reporting process in a single paramedic service in Ontario, Canada, we found that 25% of violent encounters involved sexist, misogynistic, racist, or homophobic abuse, ultimately affecting 13% of the active-duty paramedic workforce. Expressed as rates, our findings correspond to paramedics experiencing potentially hate-motivated violence every 3 days. Compared to other forms of violence, harassment on protected identity grounds was associated with an increased risk of self-reported emotional distress at the time of reporting. Our findings offer unique insight into the type of violence experienced by paramedics, but—amid a culture of under-reporting—our estimates remain conservative, and both the scope of the problem and potential harm are likely much higher.

## Figures and Tables

**Figure 1 ijerph-21-00505-f001:**
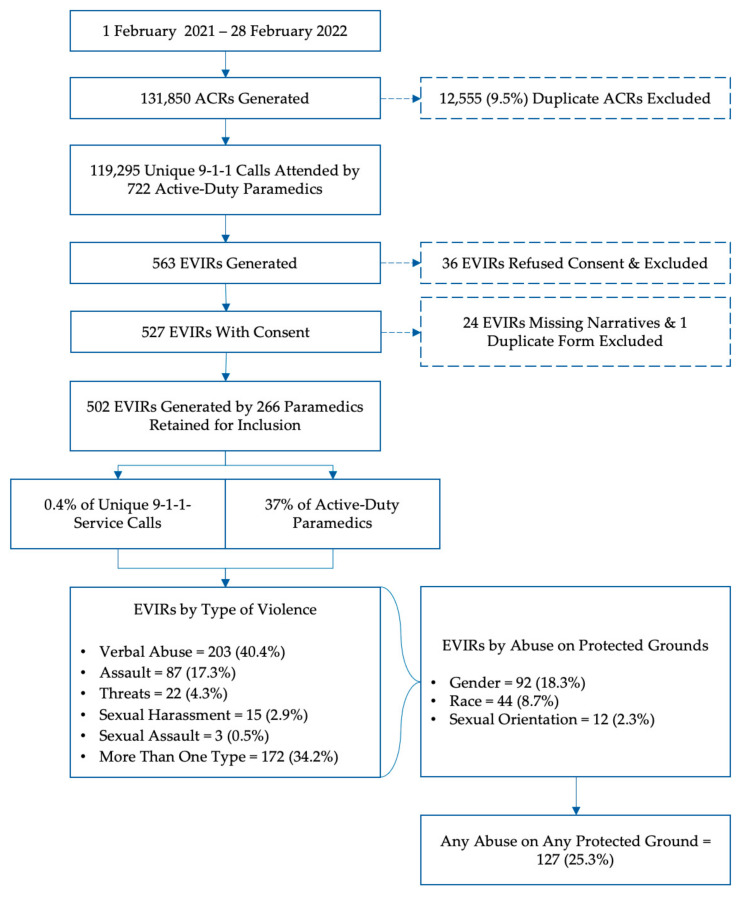
Breakdown of included cases. ACR = Ambulance Call Report, EVIR = External Violence Incident Report.

**Table 1 ijerph-21-00505-t001:** Breakdown of inter-rater agreement across domains. K = Kappa, SE = standard error, CI = confidence interval (calculated at 95%).

Domain	Rater 1	Rater 2	K	SE	Lower CI	Upper CI	*p*
Gender	91	78	0.837	0.03	0.77	0.90	<0.001
Race	49	42	0.795	0.04	0.70	0.88	<0.001
Sexual Orientation	13	9	0.721	0.10	0.50	0.93	<0.001

**Table 2 ijerph-21-00505-t002:** Percent agreement and resolution of discrepant cases. *N* = 502.

Domain	(A) Complete Agreement	(B) Discrepant Cases	(C) Resolved as ‘No’	(D) Resolved as ‘Yes’	Final N (A + D)	%
Gender	73	24	5	19	92	18%
Race	37	16	9	7	44	9%
Sexual Orientation	8	5	1	4	12	2%

## Data Availability

Data for this study may be shared with interested researchers on a case-by-case basis, subject to a privacy review and formal data sharing agreement.

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
