# Peer review of "Sexist, Racist, and Homophobic Violence against Paramedics in a Single Canadian Site"

_ijerph, 2024, doi:10.3390/ijerph21040505_

Round 1

Reviewer 1 Report

Comments and Suggestions for Authors

This is a well written, competent and clearly presented paper. I do suggest, though, that the authors give some consideration to structural/societal context. I.e. while identification of emotional distress is considered, it is also important to acknowledge broader impacts in terms of the perpetuation of inequality (gender, race, sexuality) beyond the individual. A contextual perspective also brings into focus the contexts within which abuse occurs. That is, abuse both reflects and perpetuates social divisions and inequality.

Author Response

Dear colleague,

Thank you for taking the time to review our manuscript and for your encouraging and constructive feedback. I take your point fully and we did make an effort to speak to the issue you raised in our revision; however, neither myself or the senior author are versed in this literature and I regret that we have likely not captured the essence of the important point you raised to its fullest extent. The journal gave us but five weekdays to effect a revision and declined a further extension.

Again, your point is well taken and your feedback was very encouraging. Thank you for making the time.

Justin

Reviewer 2 Report

Comments and Suggestions for Authors

Thank you for the opportunity to review the submitted paper, which is very innovative in its conception, I appreciate especially the methodological grounding and data collection for practical implementation. 

Although the approach of the authors has some limitations, which the authors themselves are aware of, as they mention them in the relevant part of the text, I consider the text to be socially and scientifically significant in its topic. I have virtually only two minor comments and recommendations - firstly, I would recommend including a section on the current state of knowledge and research and studies on a similar topic, and also referring to relevant and more recent literature (e.g. line 67 contains a reference to literature from 2014, with a disclaimer that the data is available for the most recent year, but logically this is 2013, which is not only confusing, but there are more recent sources proving what the authors are trying to convey. 

Further, the authors state in line 117 that paramedics can document a detailed narrative description. But what happens if they don't, i.e. if it is not an obligation? To what extent might this have affected the data viewed? 

I would recommend the conclusion to be more elaborated, given the importance of the text and its interest not only for the academic community, it is rather abbreviated. 

Author Response

Comment

Response

Page/Line #

I would recommend including a section on the current state of knowledge and research and studies on a similar topic, and also referring to relevant and more recent literature

I had hoped to address this more fully with a more nuanced review of the literature that we cite in our manuscript; however, the journal offered an initial timeline of but five weekdays to effect – what they described as – minor revisions. Expanding the scope and detail of the literature review was out of scope for a minor revision.

N/A

Further, the authors state in line 117 that paramedics can document a detailed narrative description. But what happens if they don't, i.e. if it is not an obligation? To what extent might this have affected the data viewed?

You are quite right that missing data can complicate the analysis – in the results section and in Figure 1, we explain that 24 reports (out of the initial pool of 563; 4.2%) were missing the narrative description field altogether. These reports were excluded. I suspect it would bias our results toward underestimation by an unknown degree. We do explain in the limitations and conclusion that there are various factors that nudge our findings toward underestimation of the true prevalence. Between missing data and persistent underreporting, the latter likely has the greater effect

N/A

I would recommend the conclusion to be more elaborated, given the importance of the text and its interest not only for the academic community, it is rather abbreviated

Was there something specific you were looking for that may have been overlooked? I sat with this for quite some time reviewing the conclusion as written; I agree it is brief, but it gives (1) the prevalence; (2) an estimation of rate; (3) an assessment of the potential impact on the well-being of the paramedics, while (4) emphasizing that the true scope and potential harm are likely higher. It is blueprinted to our study’s two objectives and reflective of the data we have. It’s not that I want to be disagreeable, I’m just cognizant that this is a complex topic, and I don’t want to stray too far from the purpose of the study and the data we have.

N/A

Reviewer 3 Report

Comments and Suggestions for Authors

1. In the background it has not been explained what the contribution of this study is to the development of science and technology.

2. In the background, the novelty of this study has not been explained compared to previous studies that have been conducted.

3. The method does not explain how the study participants were selected. is it random?

4. In the method it is not explained whether the participants in this study have agreed to the information of concern.

5. Does this study use inclusion and exclusion criteria? please mention clearly

6. One of the variables in this study is emotional distress. How to measure emotional distress? Is it valid to only use 1 question with a "yes" or "no" answer choice? What is the validity and reliability of the questionnaire?

7. Table 1 is still difficult to understand because in 1 table there are the same domain categories "Gender, Racist, but with different columns. Wouldn't it be better to just separate the top and bottom tables?

8. In line 195 it says N=96, but in Table 1, we don't see that number, either in terms of N=96 or in terms of percentage N=100%

9. In Figure 1 it is stated that Gender=92 (18.3%). Where does the 18.3% percentage come from? even though it was explained that the participants who were suitable for inclusion were 266 paramedics

10. Please check again the actual population and sample numbers in this study. What are the inclusion criteria? how to choose the participants!

11. Please use the latest references, especially references number 4, 9, 27, 38, 39

Author Response

Comment

Response

Page/Line #

In the background it has not been explained what the contribution of this study is to the development of science and technology.

Our findings don’t contribute much to technology, but we do believe they contribute meaningfully to the body of research on violence against paramedics. The logic is thus: violence is underreported ® underreporting creates a gap in research ® what research does exists mostly uses surveys ® surveys lack granularity and specificity and don’t capture detailed event level data about violent encounters, so non-physical violence (like racial slurs) will be missed ® our (novel) reporting process captures detailed, event-level data about violent encounters, broadly defined ® we can now estimate how commonly paramedics encounter non-physical violence ® this is a novel contribution to research. We do explain this in the introduction and the opening arguments are carefully signposted; however, we have added some additional wording in the relevant sections to ensure we use the words “gap” and “contribution” and “novel”.

Page 3, “Non-Physical Violence”, Page 4, “The External Violence Incident Report”

In the background, the novelty of this study has not been explained compared to previous studies that have been conducted.

There’s some overlap here with how we responded above. We have tried to make the gap, contribution, and novelty more explicit.

As above

The method does not explain how the study participants were selected. is it random?

The study participants were included if they experienced violence. Whether this was random or targeted is difficult to say with certainty and beyond the scope of our investigation.

N/A

In the method it is not explained whether the participants in this study have agreed to the information of concern.

I’m a bit unclear on what you’re referring to here. If you mean participant consent, we do offer an explanation of the consent process in the back matter in the paragraph titled “Informed Consent Statement”

N/A

Does this study use inclusion and exclusion criteria? please mention clearly.

Any paramedic who filed a violence report and consented to use of the report for research purposes was included in the study.

N/A

One of the variables in this study is emotional distress. How to measure emotional distress? Is it valid to only use 1 question with a "yes" or "no" answer choice? What is the validity and reliability of the questionnaire?

It is, unfortunately, limited to just the one question on the report. The question asks paramedics if they were ‘emotionally impacted’ and they can answer ‘yes’, ‘no’, or ‘I’m uncertain’. The reasons the question was set up this way are complicated, and I don’t love it either, but it’s the best we’ve got.

N/A

Table 1 is still difficult to understand because in 1 table there are the same domain categories "Gender, Racist, but with different columns. Wouldn't it be better to just separate the top and bottom tables?

No problem, that’s a helpful suggestion. We split the table into two.

Pages 7 & 8, Tables 1 & 2, respectively.

In line 195 it says N=96, but in Table 1, we don't see that number, either in terms of N=96 or in terms of percentage N=100%

That’s correct; 96 paramedics filed a report that (our reviewers felt) described some form of sexist, misogynistic, racist, or homophobic content. This number is only reported in the main text of the paper.

N/A

In Figure 1 it is stated that Gender=92 (18.3%). Where does the 18.3% percentage come from? even though it was explained that the participants who were suitable for inclusion were 266 paramedics

It is 92 out of the 502 EVIRs included in the study (92/502 = 18.3%).

N/A

Please check again the actual population and sample numbers in this study. What are the inclusion criteria? how to choose the participants!

I checked them again; everything is fine. There are no inclusion criteria per se, as we had rather little control over which paramedics were subjected to violence.

N/A

Please use the latest references, especially references number 4, 9, 27, 38, 39

I checked references 4, 9, 27, 38, and 39 and these are the most up-to-date versions of the bibliographic details for each.

N/A